# Comparative Evaluation of Flexural Strength of Four Different Types of Provisional Restoration Materials: An In Vitro Pilot Study

**DOI:** 10.3390/children10020380

**Published:** 2023-02-15

**Authors:** Hafsa Al Idrissi, Lovely Muthiah Annamma, Dalya Sharaf, Ahmad Al Jaghsi, Huda Abutayyem

**Affiliations:** 1Specialist Prosthodontist Hospital Dr. Rami Hamed Center, Building 52 Dubai Healthcare City, Dubai P.O. Box 212619, United Arab Emirates; 2Department of Clinical Sciences, College of Dentistry, Ajman University, Ajman P.O. Box 346, United Arab Emirates; 3Center of Medical and Bio-Allied Health Sciences Research, Ajman University, Ajman P.O. Box 346, United Arab Emirates; 4College of Dentistry, Ajman University, Ajman P.O. Box 346, United Arab Emirates; 5Department of Prosthodontics, Gerodontology and Dental Materials, Greifswald University Medicine, 17489 Greifswald, Germany

**Keywords:** provisional restoration, flexure strength, bis-acryl composite resin, PMMA, urethane dimethacrylate resin

## Abstract

With provisional restorations, properties such as flexural strength play a key role in maintaining the abutment teeth in position over the interim period until the final restorations are placed. This study aimed to evaluate and compare the flexural strength of four commonly used provisional resin materials. Ten identical 25 × 2 × 2 mm specimens were made from four different groups of provisional resin material, namely 1: SR Ivocron (Ivoclar Vivadent) cold-polymerized poly-methyl methacrylate (PMMA), 2: S Ivocron (Ivoclar Vivadent) heat-polymerized PMMA, 3: Protemp (3M Germany-ESPE) auto-polymerized bis-acryl composite, and 4: Revotek LC (GC Corp, Tokyo) light-polymerized urethane dimethacrylate resin. The mean values of the flexural strength of each group were calculated and the data were analyzed using one-way ANOVA and Tukey post hoc tests. The mean values (MPa) were as follows: for cold-polymerized PMMA, it was 125.90 MPa; for heat-polymerized PMMA, it was 140.00 MPa, with auto-polymerized bis-acryl composite 133.00 MPa; and for light-polymerized urethane dimethacrylate resin, it was 80.84 MPa. Thus, the highest flexural strength was recorded with heat-polymerized PMMA and the lowest flexural strength with light-polymerized urethane dimethacrylate resin, which was significantly low. The study did not detect a significant difference in the flexural strengths of cold PMMA, hot PMMA, and auto bis-acryl composite.

## 1. Introduction

Provisional restorative materials are interim materials used to stabilize the prepared teeth when in use, or to protect them between treatment sessions. Interim restorations are placed on the prepared teeth of inlays, onlays, single crown restorations, fixed partial dentures, and implant abutments. The temporary restoration prevents damage or fracture of the abutment teeth while permanent restorations are being fabricated. To be successful, these restorations should fulfill biological, mechanical, and esthetic requirements [1].

We were unable to identify, from the literature, any single material that fulfills all the criteria for an ideal provisional restoration. These include good physical properties and marginal adaptation, as well as adequate retention and resistance to dislodgement during tooth function. The material should be strong, durable, hard, and nonirritating to pulp and other tissues. An ideal material should possess low exothermicity and should be nonporous and dimensionally stable. Anterior provisional restoration materials should be esthetic and color-stable with a high polish. Such materials should be inexpensive and easy to manipulate, remove, and re-cement by the dentist [2]. To meet these requirements, the provisional restoration material should have specific mechanical and physical properties such as good flexural strength, increased wear resistance, dimensional stability, minimal marginal gap formation, and increased resistance to staining and discoloration [3].

There are various categories of provisional material, including chemically activated auto-polymerized acrylic resins, heat-activated acrylic resins, light-activated acrylic resins, and dual light- and chemically-activated acrylic resins. Recently, temporary CAD CAM materials have been used with success. Some examples are VITA CAD-Temp^®^, Polyetheretherketone “PEEK”, and Telio CAD-Temp [4]. However, long-span bridges and restorations for patients with parafunctional habits are considered challenging and demanding, particularly for interim restorations. Unless the strength of these materials is good, they will not be able to withstand the load of occlusal forces [5].

Acrylic resins are the most common materials used for custom provisional restorations [6,7]. Though acrylic resins are brittle, their greatest advantage is the ease with which they can be altered by additions and subtractions [8,9]. 

The most commonly used monomers are Methyl methacrylate, Ethyl methacrylate, Isobutyl methacrylate, Bisphenol A diglycidylether methacrylate (bis-GMA), and Urethane dimethacrylate [10]. The different monomers possess different physical and mechanical properties. The molecular structure of the alkyl group in PMMA gives it higher strength in comparison to PEMA [11]. 

According to Yannikakis, a period of 30 days is considered a long-term provisionalization stage [12]. When temporary crowns are used for long periods, there is a higher risk of discoloration. Heat-polymerized acrylic resins are denser, stronger, more wear-resistant, color-stable, and resistant to fracture than auto-polymerized acrylic resins [13]. Therefore, a heat-polymerized provisional restorative resin is recommended [12].

This study aimed to evaluate and compare the flexural strength of four provisional resin materials from four different groups. The null hypothesis for this study was that there would be no difference between the flexural strength of the four types of provisional dental materials.

## 2. Materials and Methods

Four groups of ten specimens each were prepared from the provisional resin material listed in Table 1. Initially, 15 specimens were prepared from the same batch for each group. On observation under the stereomicroscope, any sample with minor defects or size variation was discarded. 

### 2.1. Methodology: (Figure 1)

A customized mold was fabricated using a putty index. The mold was packed in a brass flask as a standard mold for all tested materials (Figure 2a). The dimensions of the mold were 40 × 45 × 15 mm. Each of the four provisional restorative materials to be tested was initially fabricated in the dimensions of 40 × 45 × 15 mm. (Figure 2a). 

**Figure 1 children-10-00380-f001:**
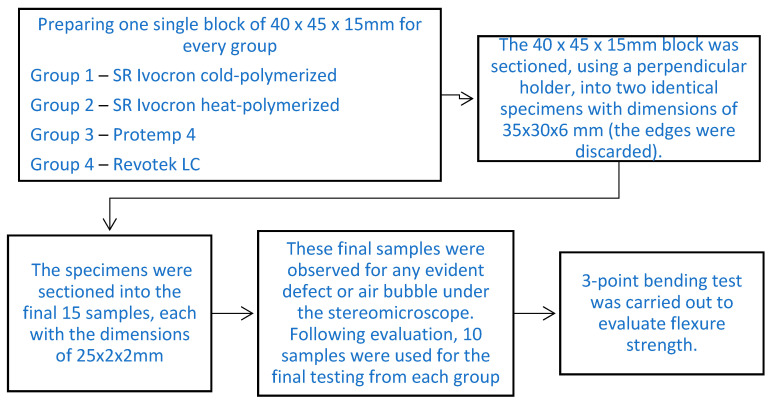
Sample preparation flow chart.

The materials were numbered as follows: Group 1–SR Ivocron cold-polymerized, Group 2–SR Ivocron heat-polymerized, Group 3–Protemp 4, and Group 4–Revotek LC. 

#### 2.1.1. Group 1–SR Ivocron Cold-Polymerized PMMA

The material for group 1 is supplied as polymer and monomer. The manipulation of the material was carried out in a rubber dappen dish. The monomer in liquid (SR Ivocron, Ivoclar Vivadent, Schaan, Liechtenstein) and powder (Dentine 130, SR Ivocron, Ivoclar Vivadent) form were added to the dappen dish according to the manufacturer’s instructions. The material was mixed until all the polymer particles were thoroughly wetted with the monomer, with the monomer and polymer forming a homogenous mix. Once the material reached the dough-like stage, it was packed into the mold within the brass flask and closed under intermitted pressure using a bench press (Hydraulic press, Press, Dentalfarm, Italy). The flask was then put in a pressure pot filled with hot water (TUV, Dental lab pressure pot, Germany) with a pressure of 2 bars and kept there for 15 min to ensure complete polymerization. The specimen was then retrieved (Figure 2a,b).

#### 2.1.2. Group–2 SR Ivocron Heat-Polymerized PMMA

The material for this group is also supplied as polymer and monomer. The manipulation of the material was carried out in a rubber dappen dish. The polymer powder (Dentine 130, SR Ivocron, Ivoclar Vivadent) and the Press-liquid monomer (SR Ivocron, Ivoclar Vivadent) were added according to the manufacturer’s instructions. The material was mixed vigorously until all the polymer particles were thoroughly wetted with the monomer. It was then covered for 15 min, following the manufacturer’s instructions. Once it reached a dough-like stage, the material was packed into the mold within the brass flask. The flask was then closed and kept under intermittent pressure in a bench press (Hydraulic press, Press, Dentalfarm, Italy). Trial closure was performed until all the excess material was removed. The flask was then placed in warm water in an acrylizer (preheated at 70 degrees Celsius), boiled for 30 min and finally cooled gradually. The flask was allowed to cool for 45 min and the specimen was retrieved (Figure 2a,b).

#### 2.1.3. Group–3 Protemp 4 

The material was dispensed using the disposable tips provided by the manufacturers into the mold lubricated with petroleum jelly. The flask was then closed using intermittent pressure in a bench press (Hydraulic press, Dentalfarm, Italy). After eight minutes, the specimen was then retrieved from the flask (Figure 2a,b).

#### 2.1.4. Group 4–Revotek LC

The flask was filled with the material as per the manufacturer’s instructions, using the provided spatula. The filled flask was closed and then kept under pressure using a bench press (Hydraulic press, Press, Dentalfarm, Italy) for trial closure. On reopening the flask, the excess material was removed, and a glass slab was used to cover the surface of the provisional material. A powered visible light curing machine (Eclipse junior resin system, Dentsply) was used to ensure full polymerization of the material, for 15 min per side. The specimen was then retrieved from the flask (Figure 2a,b).

#### 2.1.5. Preparing Final Specimens for Testing

A standardized procedure was used for constructing the specimens of all four different blocks/groups.

The blocks were inspected for defects such as air bubbles or external voids by visual examination. The blocks were sectioned with the Isomet low-speed saw (Buehler, 11-1280-160, USA), wafering diamond blade 102 mm diameter × 0.3 mm (Buehler, 11-4254, series 15 LC diamond, USA), and lubricant oil (Buehler, Isocut fluid, 11-1193-032, 11-1193-128, USA). The 40 × 45 × 15 mm block was sectioned using a perpendicular holder into two identical specimens, each with the dimensions of 35 × 30 × 6 mm, as shown in Figure 3. 

The edges of each block were discarded, while the remaining portion of the rectangular specimens was sectioned further with the help of the Isomet cutting machine calipers. The caliper dimension was adjusted to 2.4 mm, taking into consideration the disc thickness (every two sections from the starting point measured 1 mm in thickness). The specimens were sectioned to obtain the final 15 samples of each of the four provisional restorative materials, Figure 4. The final samples had dimensions of 25 × 2 × 2 mm. A total of 60 specimens were obtained from all four groups of provisional restorative materials. 

The 60 specimens were washed under running water and air-dried. The final selection included 10 per group. The specimens chosen were required to have perfect dimensions and did not show any defect under the stereomicroscope.

### 2.2. Flexure Strength Testing

Sample storage conditions before testing satisfied the ISO 10477:2004 regulations (wet medium for 24 h). The calibration of the universal testing machine was performed, to achieve a crosshead speed of 0.5 mm per minute with a 5 kN load cell.

The specimens were tested for flexural strength using the 3-point bending test. The test was conducted using equipment from (Model M350-5CT; Testomatric, Rochdale, UK.) The load was applied with a 2 mm diameter sphere to the center of the specimen. The specimens were loaded till failure. The load at fracture was recorded for each group (Figure 5).

## 3. Results

The mean flexural strength of the groups were recorded (Figure 6). The highest flexural strength was demonstrated by Group 2–SR Ivocron heat-polymerized PMMA (140.0 MPa), followed by Group 3–Protemp 4 (133.18 MPa), and then Group 1–SR Ivocron cold-polymerized PMMA (125.91 MPa). The lowest flexural strength was recorded by Group 4–Revotek (80.85 MPa), which was significantly lower (*p*-value = 0.001) than those recorded with groups 1, 2, and 3 (Figure 6, and Table 2). However, the study could not detect a significant difference in flexural strength between the first three groups. 

### Statistical Analysis

The obtained data were tabulated and statistically analyzed using (IBM SPSS statistics version 28.0.0.0) with a significant level fixed at 5% (α = 0.05). The data were analyzed by one-way ANOVA and the post hoc test. (Table 2 and Table 3). One-way ANOVA was used to compare the mean flexural strengths of each of the four groups. The statistical significance was set at a *p*-value less than 0.05. ANOVA results show significant differences between the mean flexural strengths of the four groups, Table 2. Hence post hoc testing was applied to reveal the differences between the means of two groups, when the groups were compared by pairs (Table 3). 

The post hoc test indicated a significant difference between Revotek LC (Group 4) and the other three groups of the tested provisional resin materials (*p* < 0.05). There was no significant difference in the mean flexural strength of the other groups: Protemp 4 auto-polymerized bis-acryl composite (Group 3), SR Ivocron heat-polymerized PMMA (Group 2), and SR Ivocron cold-polymerized PMMA (Group 1), revealed no significant differences. The results of the analysis indicate that the hot-polymerized PMMA resin showed the greatest flexural strength, followed by the auto-polymerized bis-acryl composite and then the cold-polymerized PMMA. The lowest flexural strength was observed with the light-polymerized urethane dimethacrylate resin. 

## 4. Discussion

Flexural strength is one of the key factors that affect the longevity of any restoration. At very low flexural strengths, the provisional restorations may fracture. Therefore, in this study, the most commonly used provisional restoration materials were evaluated for their flexural strength. Provisional restorations can be manufactured through the digital or conventional approach. 

Digital methods include additive (3D printing) and subtractive (milling) manufacture. The conventional approaches include two distinct methods: direct use in the oral cavity (light-curable and cold) or indirect (hot and cold). This study targeted only the conventional method, this being the most frequently used at present. 

The results of this study show that light-cured urethane dimethacrylate (Revotek) had a flexural strength that was significantly lower than the other three tested materials. The study was unable to detect a significant difference between the first three groups: the two polymethyl methacrylate groups (SR Ivocron^®^ PMMA hot-polymerized and SR Ivocron^®^ PMMA cold-polymerized), and the auto-polymerized bis-acrylic composite (Protemp 4).

In general, static loading as a method to evaluate flexural strength does not simulate the intraoral conditions, but it does provide a means to compare materials under controlled conditions. Static flexural loading studies can, to some extent, predict the performance of a material used for intraoral restorations. The fracture resistance of materials used for interim restorations also depends on the structure of the restoration and the aging processes in intraoral conditions. The average chewing force of human beings is 35 to 70 N at a frequency of 1066 Hz and the temperatures in the mouth can range between −8 °C and +81 °C. This indirectly leads to temperatures between 5 °C and 55 °C on the surfaces of restorations exposed to oral conditions [14]. Many authors have proved that the flexural strength of provisional resin materials is also influenced by exposure to saliva, food components, and beverages [15,16].

In this study, the experimental conditions were standardized, while potential errors in testing were reduced by fabricating all provisional restoration material specimens on the same day. A single trained operator handled the preparation of all the samples, and the measurements were made after prior calibration. All samples were refrigerated at 5 °C. The reason for differences in the flexural strengths between groups could probably be traced to the differences in polymerization technique, filler, and monomer content. When evaluating the flexural strength, the groups of SR Ivocron heat-polymerized PMMA (Group 2) showed the highest flexural strength among all the groups. This result is consistent with a study done by Donovan et al. [17]. In their study on the longevity of resin materials, they compared the porosity and hardness of resins polymerized under different polymerization conditions, as in air, under water, under air pressure, and water and air pressure only. They observed that polymerization within a pressure vessel using air and water increased the strength and reduced the porosity of the resin material [17]. Conventional methacrylate resins are monofunctional, linear molecules with low molecular weights, decreased strength, and rigidity. Many authors have proved that heat-polymerized resins are denser, stronger, more wear-resistant, fracture-resistant, and more color-stable than auto-polymerized resins. Moreover, improper polymerization techniques that are not carried out under adequate pressure or temperature conditions may lead to reduced material strength due to air bubble entrapment [18] In some specific cases, heat-polymerized resin may even be suitable for long-term temporary restorations [13,19]. 

The other provisional restoration materials, namely, bis-acrylic resins, were evaluated for their flexural strength and compared with methacrylate base resins by Haselton. They compared flexural strength before and after immersing in artificial saliva for 10 days. Mixed results were obtained: some samples showed good strength, and others had lower flexural strength than traditional methacrylate resins [11]. Bis-acrylic provisional materials became popular due to the convenience offered by their availability as cartridge systems. This system promoted ease of dispensing of the material, while ensuring a consistently accurate mix [20]. 

In another study, it was shown that the incorporation of multifunctional monomers (Bis-GMA or TEGDMA) into bis-acryl resin followed by reinforcement with inorganic fillers led to a composite material with increased strength and microhardness due to its cross-linking with monomers [5,21]. In view of these results, the widely used bis-acrylic composite Protemp 4 was also included in this study. The bis-acrylic composite did not differ significantly from PMMA, the gold standard among temporary restoration materials. The older version of the same brand of provisional restoration material (Protemp 3 Garant) has an acceptable flexural strength of 115.7 MPa (SD 5.7), as reported by Haselton [19]. However, it is lower than the mean flexural strength for the new Protemp 4 at 133.18 MPa (SD 7.9), as recorded in the current study. 

Materials that can be cured by visible light, or visible light cure materials (VLC), were first introduced in the 1980s. These materials have been shown to possess improved physical properties such as reduced polymerization shrinkage when incorporated with microfine silica [22,23,24]. In the current study, the VLC material Revotek LC, urethane dimethacrylate (Group 4), had the lowest flexural strength among all groups. Many authors have studied visible light-polymerized materials and observed that they produce less exothermic heat while setting, making them more pulp-friendly and providing extended working time when compared with PMMA or PEMA (Polyethyl methacrylate) [25,26,27]. Bis-acrylic materials have a less exothermic setting reaction compared to methyl methacrylates, which, coupled with their low shrinkage and good marginal adaptation, gives them an advantage over the latter [28,29]. 

Apart from the convenience of dispensing, the bis-acrylates are hydrophobic, unlike PMMA which tends to absorb water [15]. Revotek^TM^ has only 10–15% of the filler content (composed of crystalline silica powder) compared to the higher filler content in PMMA and bis-acrylates, with filler content making up above 25% in the latter. This could be the major reason for the decreased flexural strength observed with urethane dimethacrylate provisional restoration material in this study [30]. 

Flexural strength is one of several factors influencing the success of a provisional /interim prosthesis. Therefore, it is important to report the clinical behavior and failure rates and profiles of these materials. Some studies have suggested that fracture is the most important reason for the clinical failure of a provisional restoration [14]. The application of provisional restoration in children is well documented by a study conducted by Vignesh et al in which they used the strip crown resin technique to restore deciduous incisors with Pedo shade packable composite or Protemp. It was observed that the Protemp fracture strength was significantly higher than that of the Pedo shade packable composites [31]. Many authors have also reported the use of acrylic crowns for primary dentition [32]. In recent years the newly developed CAD CAM temporary crowns have been researched as a replacement for conventional provisional crown materials. They are more expensive yet less time-consuming and offer better marginal adaptation. Abdullah et al studied three different types of CAD CAM temporary crowns; VITA CAD-Temp, Polyetheretherketone, and Telio CAD Temp, and compared them to Protemp 4. They observed that CADCAM crowns gave a better marginal fit but the internal fit was more uniform with Protemp 4. They also found that the fracture strength was higher with Protemp 4 compared to CAD CAM temporary crowns [4]. 

The limitations of this study include the determination of flexural strength for provisional restoration materials in conditions that do not simulate the intraoral environment and do not include thermocycling. More studies with a bigger sample size are needed to compare the flexural strength of various CAD CAM temporary crowns materials with the conventional types after aging. Flexural strength was the mechanical property considered in the current study as it reflects partially and indirectly the tensile and compressive strength as well as the elastic modulus of a material. However, future studies should evaluate other physical properties of these provisional restoration dental materials.

## 5. Conclusions

Within the limitations of this study, the following conclusions were observed: 

The first three groups, SR Ivocron heat-polymerized PMMA, SR Ivocron cold-polymerized PMMA, and Protemp 4, had similar flexural strengths within acceptable limits. The highest flexural strength was demonstrated with heat-polymerized PMMA. The VLC urethane dimethacrylate (Revotek LC) revealed the lowest flexural strength in this study, which was significantly lower than with all other materials. Therefore, it is not the best choice for long-span bridges or as a long-term temporary restoration. However, it is more pulp-friendly as no exothermic heat is released, making it suitable for short-term temporary use. 

## Figures and Tables

**Figure 2 children-10-00380-f002:**
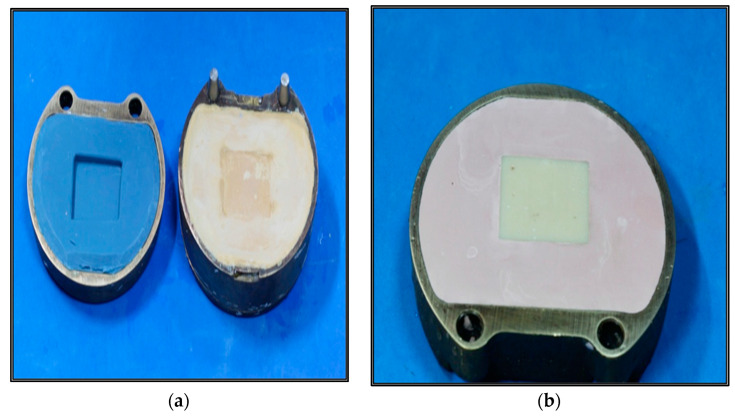
(**a**) Customized standard mold. (**b**) Fabricated resin material of 40 × 45 × 15 mm block.

**Figure 3 children-10-00380-f003:**
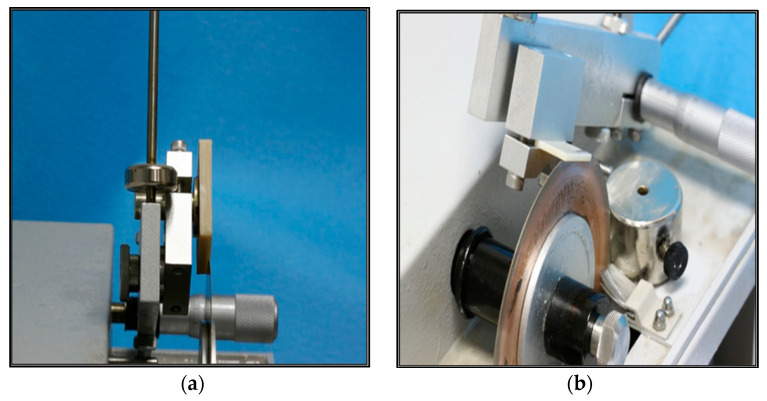
(**a**) Isomet low-speed parallel cut. (**b**) Isomet low-speed perpendicular cut.

**Figure 4 children-10-00380-f004:**
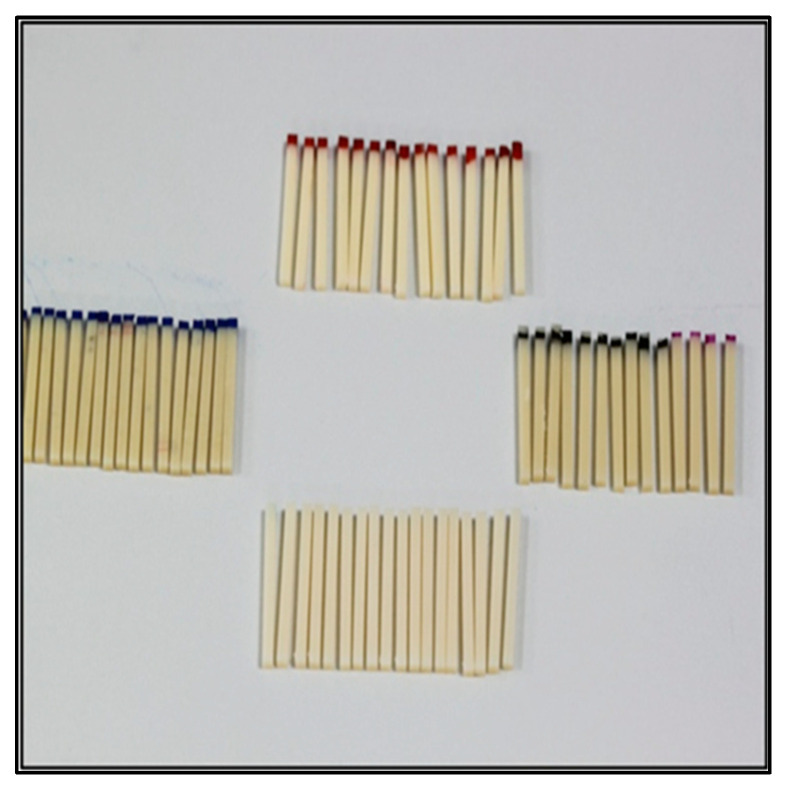
Tested specimens in all four groups.

**Figure 5 children-10-00380-f005:**
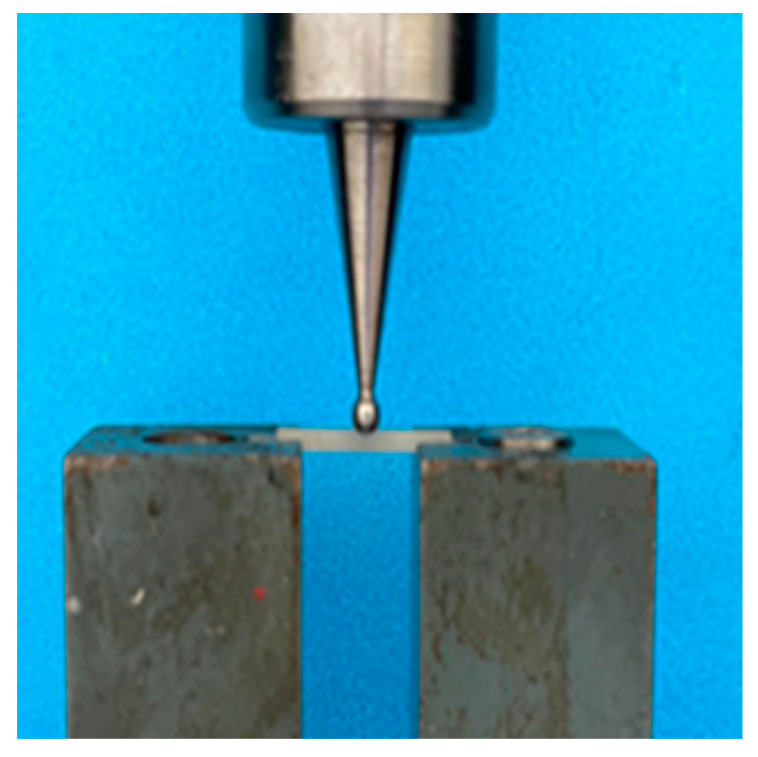
Placement of sample in UTM machine.

**Figure 6 children-10-00380-f006:**
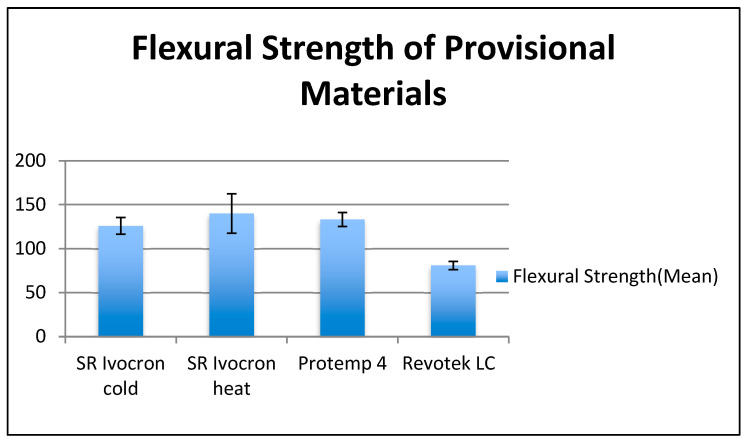
Histogram of mean flexural strength of four tested groups.

**Table 1 children-10-00380-t001:** Materials tested: The four different types of provisional restoration materials and their manufacturer.

Group	Product Name	Manufacturer	Resin Type
1	SR Ivocron	Ivoclar Vivadent, Liechtenstein	Cold-polymerized polymethyl methacrylate
2	SR Ivocron	Ivoclar Vivadent, Liechtenstein	Heat-polymerized polymethyl methacrylate
3	Protemp 4	3M Germany-ESPE	Auto-polymerized bis-acryl composite
4	Revotek LC	GC Corp, Tokyo	Light-polymerized urethane dimethacrylate resin

**Table 2 children-10-00380-t002:** Analysis using post hoc tests (Tukey).

					95% Confidence Interval		
(I) Group	(J) Group	Mean Difference (I−J)	Std. Error	Sig.	Lower Bound	Upper Bound	Remark
Cold	Heat	−14.09556	6.11505	0.118	−30.6634	2.4723	NS
	Protemp	−7.27667	6.11505	0.637	−23.8446	9.2912	NS
	Revotek	45.05778 *	6.11505	<0.01	28.4899	61.6257	S
Heat	Cold	14.09556	6.11505	0.118	−2.4723	30.6634	NS
	Protemp	6.81889	6.11505	0.683	−9.7490	23.3868	NS
	Revotek	59.15333 *	6.11505	<0.01	42.5854	75.7212	S
Protemp	Cold	7.27667	6.11505	0.637	−9.2912	23.8446	NS
	Heat	−6.81889	6.11505	0.683	−23.3868	9.7490	NS
	Revotek	52.33444 *	6.11505	<0.01	35.7666	68.9023	S
Revotek	Cold	−45.05778 *	6.11505	<0.01	−61.6257	−28.4899	S
	Heat	−59.15333 *	6.11505	<0.01	−75.7212	−42.5854	S
	Protemp	−52.33444 *	6.11505	<0.01	−68.9023	−35.7666	S

* The mean difference is significant at the 0.05 level. *S: Significant NS: Not Significant.*

**Table 3 children-10-00380-t003:** One-way ANOVA.

	Sum of Squares	df	Mean Square	F	Sig.
Between Groups	19,274.279	3	6424.760	38.181	<0.01
Within Groups	5384.720	32	168.272		
Total	24,658.998	35			

## Data Availability

All the data regarding the study are with the main author and will be shared if requested.

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
