# Peer review of "Comparative Evaluation of Flexural Strength of Four Different Types of Provisional Restoration Materials: An In Vitro Pilot Study"

_children, 2023, doi:10.3390/children10020380_

Round 1

Reviewer 1 Report

This study focused on investigating the mechanical properties of  conventional crown and FPD materials, even though new CAD/CAM milled and 3D printed technologies have surfaced in the past few years. The authors should address the following points to improve the quality of the manuscript:

- The abstract should be non-structured with specific word limit (please see the authors' guidelines).

- The authors should add a flow chart of the study.

- It is understood that this study has a future extension to include the new CAD/CAM technologies, however the significance of studying these conventional materials alone is not unique.

- The conclusion section should be expanded to include all the possible outcomes.

Author Response

We thank you for allowing us to revise and resubmit with the suggested modifications by the reviewers. We like to thank the reviewers for their valuable comments that helped us to re-edit the manuscript for better clarity. All the modified areas are highlighted in blue within the manuscript.

Reviewer 1

This study focused on investigating the mechanical properties of the conventional crown and FPD materials, even though new CAD/CAM milled, and 3D printed technologies have surfaced in the past few years. The authors should address the following points to improve the quality of the manuscript:

Thank you for your valid points, we are in total agreement with you concerning adding the CAD CAM provisional materials as part of the study. However, the main reason for this pilot study was to evaluate the materials used in under and postgraduate programs at our institution.

Secondly, we are going to include in our extended coming study different types of long-term temporary CAD CAM and 3D printed materials as a continuation study.

The abstract should be non-structured with a specific word limit (please see the authors' guidelines).

The abstract is adjusted and modified as nonstructured and the changes are highlighted in blue in the manuscript.

The authors should add a flow chart of the study.

  1. Material manipulation and sample preparation.
  2. A flow chart for study design is added.

Figure 1 Samples preparation flow chart.

 It is understood that this study has a future extension to include the new CAD/CAM technologies, however, the significance of studying these conventional materials alone is not unique.

 Thank you for your valid comments. The evaluation of new CAD CAM materials will be conducted soon as a continuation of this study. Moreover, this study aimed to evaluate the commonest materials used in Undergraduate and Master programs at our university.

The conclusion section should be expanded to include all the possible outcomes.

The conclusion is expanded.

Within the limitations of this study, the following conclusions were observed:

The first three groups, SR Ivocron heat polymerizing, SR Ivocron cold polymerizing, and Protemp4 had similar flexural strengths within acceptable limits. However, heat-polymerized PMMA had the highest flexural strength. VLC urethane dimethacrylate Revotek LC group revealed the lowest flexural strength in this study. It was significantly lower than all other materials. Therefore, it is not the best choice for long-span bridges or as a long-term temporary restoration. However, it is more pulp friendly as no exothermic heat is released hence for short-term temporary use, this material can be utilized.

Reviewer 2 Report

What is the rationale for choosing these specific materials for comparison, and how do they represent a range of commonly used provisional restoration materials?

Were any precautions taken to minimize the influence of other factors that might affect the flexural strength of the specimens, such as surface roughness or moisture absorption?

Is the sample size of 10 specimens per group sufficient to provide reliable results for the comparison of the four materials?

The conclusion should be modified to highlight the key findings.

The limitations of the study should be acknowledged and discussed.

Author Response

We thank you for allowing us to revise and resubmit with the suggested modifications by the reviewers. We like to thank the reviewers for their valuable comments that helped us to re-edit the manuscript for better clarity. All the modified areas are highlighted in blue within the manuscript.

Reviewer 2

What is the rationale for choosing these specific materials for comparison, and how do they represent a range of commonly used provisional restoration materials?

Thank you for your valid points, we totally agree with you. The main reason for this pilot study is to evaluate the materials used for provisional restoration in under- and postgraduate programs at the University.

Secondly, we are going to include in our extended study different types of long-term temporary CAD CAM and 3D printed materials as a continuation study.

Were any precautions taken to minimize the influence of other factors that might affect the flexural strength of the specimens, such as surface roughness or moisture absorption?

To ensure standardization during the specimen preparation, samples for every material were prepared on the same day and stored in the same place with a calibrated single examiner. Initially, 60 specimens were prepared and after evaluation under stereomicroscope samples with altered surface roughness or minor defect and air bubbles were eliminated.

Sample storage was followed as per regulations of ISO 10477:2004 (wet medium for 24 hours) before testing. The samples were prepared from the same batch of materials.

Is the sample size of 10 specimens per group sufficient to provide reliable results for the comparison of the four materials?

Previous studies used data to calculate the minimum sample size (10 for every group). (Debra R. Haselton, Ana M. Diaz-Arnold, Marcos A. Vargas, Flexural strength of provisional crown and fixed partial denture resins, The Journal of Prosthetic Dentistry, Volume 87, Issue 2, 2002).

   The study was carried out as a pilot study with 15 specimens. The specimens with defects or surface roughness detected through the stereomicroscope were excluded; therefore,  the final sample size was reduced to 10 for every group. Further studies with bigger sample sizes will be carried out in the future, including the available CAD CAM material too.

The conclusion should be modified to highlight the key findings.

The conclusion is modified

Within the limitations of this study, the following conclusions were observed:

The first three groups, SR Ivocron heat polymerizing, SR Ivocron cold polymerizing, and Protemp4 had similar flexural strengths within acceptable limits. However, heat-polymerized PMMA had the highest flexural strength. VLC urethane dimethacrylate Revotek LC group revealed the lowest flexural strength in this study. It was significantly lower than all other materials. Therefore, it is not the best choice for long-span bridges or as a long-term temporary restoration. However, it is more pulp friendly as no exothermic heat is released hence for short-term temporary use, this material can be utilized.

The limitations of the study should be acknowledged and discussed.

The discussion and limitations were added and modified within the manuscript.

Flexural strength is one of the factors affecting the longevity of any restoration. If the flexural strength is very low the provisional restorations can fracture. Therefore, in this study, the commonest used provisional restoration materials were evaluated for their flexural strength. Provisional restorations can be manufactured through the digital or conventional approach. The digital can be additive (milling) or subtractive (3D printing). The traditional approach includes two distinct methods: direct use in the oral cavity (light-curable and cold) or indirect (hot and cold). This study targeted only the most used approach; conventional. The study found that light-curable Urethane dimethacrylate (Revotek) flexural strength was significantly lower than the other three tested materials. The study was unable to detect a significant difference between the first three groups: Polymethyl methacrylate (SR Ivocron® PMMA hot and SR Ivocron® PMMA cold), and auto-polymerized bis-acrylic composite (Protemp 4).

In general, static loading to evaluate flexural strength does not imitate the intraoral conditions, but it is always useful to compare materials under controlled conditions. Static flexural loading studies can (to some extent) predict the performance of a material used as restorations intraorally. The fracture resistance of interim materials also depends on the structure of the restoration and the aging processes in intraoral conditions. The average chewing force of human beings is 35 to 70 N at a frequency of 1066 Hz and the temperatures in the mouth can range between -80C and +810C. This indirectly transfers temperatures on the surfaces of restorations exposed to oral conditions between 50C and 550C[14]. Many authors have proved that the flexural strength of provisional resin materials is influenced by saliva, food components, and beverages[15], [16].

Standardization of experimental conditions was maintained and potential errors in testing were reduced by fabricating all provisionals on the same day. A single trained and calibrated operator handled the sample making and the measurements. All samples were refrigerated at 5 degrees Celsius. The reason for differences in the flexure strength of each group probably could be differences in polymerization technique, filler, and monomer content. When evaluating the flexure strength, the groups of SR Ivocron heat PMMA (Group 2) showed good flexure strength compared to other groups. This result is consistent with a study done by Donovan et al[17]. In their study on the longevity of resin materials, they compared the porosity, and hardness of resins polymerized under different polymerization conditions as in air, under water, under air pressure, and water and air pressure only. They observed that polymerization with a pressure vessel using air and water increased the strength and reduced the porosity of the resin material[17]. Conventional methacrylate resins are monofunctional, linear molecules with low molecular weight, decreased strength, and rigidity. Many authors have proved that heat-polymerized resin materials are denser, stronger, wear-resistant, fracture-resistant, and color-stable than auto-polymerized resins. Moreover, improper polymerization techniques not done under adequate pressure or temperature can decrease the material strength due to air bubble entrapment[18] In some specific cases, the heat-polymerized resin can also be used for long-term temporary restorations[19],[13].

The other provisional restoration material, bis-acrylic resins were evaluated for their flexural strength and compared with methacrylate base resins by Haselton. They compared flexural strength before and after immersing in artificial saliva for 10 days. Mixed results were obtained; some samples showed good strength, and others had lower flexural strength than traditional methacrylate resins[20]. Bis acrylic provisional materials became popular due to their cartridge system. The dispensing of the material is easy and delivers an accurate consistent mix[21] In another study, it was proved that when bis-acryl resins were incorporated with multifunctional monomers (Bis-GMA or TEGDMA) and when reinforced with inorganic fillers, increased the strength and microhardness due to cross-linking with monomers[5],[22]. Therefore, the widely used bis acrylic composite (Protemp) was also included in this study. The bis-acrylic composite did not differ significantly from the gold standard temporary material (PMMA). The old version of the same brand of provisional restorations (Protemp 3 Garant) has an acceptable flexural strength of 115.7 (SD 5.7), as observed by Haselton [19]. However, it is lower than the mean flexural strength for the new Protemp 4 as recorded in the current study, 133.18 (SD 7.9).

Visible light curing materials, (VLC) were first introduced in the 1980s. These materials have been shown to improve physical properties such as reduced polymerization shrinkage when incorporated with microfine silica[23], [24], [25]. In the current study, the result of light curable Revotek LC, urethane dimethacrylate (Group 4), revealed the lowest flexural strength in comparison to other groups. Many authors have studied visible light-polymerized materials and observed that they produce less exothermic heat while setting hence they are pulp friendly and have extended working time when compared with PMMA or PEMA (Polyethyl methacrylate) [26] [27],[28]. The advantage of Bis acrylic materials over methyl methacrylates is that they have a less exothermic setting reaction with low shrinkage and good marginal adaptation[29],[30].

Apart from the convenience in the dispensation technique, the Bisacrylates are hydrophobic, unlike polymethylmethacrylate which tends to absorb water.[15]  The Revotech TM compared to Bisacrylates and PMMA materials have only 10- 15 % of the filler content (crystalline silica powder) compared to the higher filler content above 25% in Bis acrylates. This could be the major reason for the decreased flexure strength observed in Urethane dimethacrylate provisional restorations[31]

Flexural strength is one of several factors influencing the success of a provisional /interim prosthesis. Therefore reporting the material behavior and failure clinically is important. Some studies have observed fracture as the most common failure of provisional restorations clinically[14] The application of provisional restoration in children is well documented by a study conducted by Vignesh et al in which they used the strip crown resin technique to restore deciduous incisors with pedo shade packable composite and compared it with Protemp. It was observed that the Protemp fracture strength was significantly higher than the pedo packable composites.[32] Many authors have also reported the use of acrylic crowns for primary dentition[33] In recent years the newly developed CAD CAM temporary crowns have been researched as a replacement for conventional provisional crown materials. They are expensive yet less time-consuming and with better marginal adaptation. Abdullah et al studied three different types of CAD CAM temporary crowns; VITA CAD-Temp, Polyetheretherketone, and Telio CAD Temp, and compared them to Protemp 4. They observed that marginal fit is good for CADCAM crowns, but the internal fit was uniform in Protemp 4. They also found that the fracture strength was good for Protemp 4 when compared to CAD CAM temporary crowns[4].

     The limitation of this study is the absence of oral environment simulation or thermocycling. More studies are needed with a bigger sample size to compare the flexure strength of various CAD CAM temporary crowns materials with the conventional types after aging. Flexural strength was the mechanical property considered in the current study as it reflects partially and indirectly the tensile and compressive strength along with the elastic modulus of a material. However, more studies are needed to evaluate the rest of the physical properties.

Reviewer 3 Report

Good paper with correct methodology. In Reference No 14 the year of publication is missing. Of course, there are very limitations of this study:

-Absence of oral environment simulation or thermocycling

-small samples size

-abcence of newest various CAD-CAM temporary crowns materials

Author Response

Thank you for the suggestions.

Limitations of the study have been highlighted in the manuscript and added

The limitations of this study include the determination of flexural strength for provisional restoration materials in conditions that do not simulate the intraoral environment and do not include thermocycling. More studies with a bigger sample size are needed to compare the flexural strength of various CAD CAM temporary crowns materials with the conventional types after aging. Flexural strength was the mechanical property considered in the current study as it reflects partially and indirectly the tensile and compressive strength as well as the elastic modulus of a material. However, future studies should evaluate other physical properties of these provisional restoration dental materials.

Round 2

Reviewer 2 Report

Overall it is a good paper

Author Response

Thank you very much for your suggestions.

English editing has been done and track changes have been activated
